# IMPROVING RELATIONAL REGULARIZED AUTOENCODERS WITH SPHERICAL SLICED FUSED GROMOV WASSERSTEIN

**Khai Nguyen**
VinAI Research, Vietnam
v.khainb@vinai.io

**Son Nguyen**
VinAI Research, Vietnam
v.son3@vinai.io

**Nhat Ho**[*]
University of Texas, Austin
VinAI Research, Vietnam
minhnhat@utexas.edu

**Tung Pham**
VinAI Research, Vietnam
v.tungph4@vinai.io

**Hung Bui**
VinAI Research, Vietnam
v.hungbh1@vinai.io

## ABSTRACT

Relational regularized autoencoder (RAE) is a framework to learn the distribution of data by minimizing a reconstruction loss together with a relational regularization on the latent space. A recent attempt to reduce the inner discrepancy between the prior and aggregated posterior distributions is to incorporate sliced fused Gromov-Wasserstein (SFG) between these distributions. That approach has a weakness since it treats every slicing direction similarly, meanwhile several directions are not useful for the discriminative task. To improve the discrepancy and consequently the relational regularization, we propose a new relational discrepancy, named *spherical sliced fused Gromov Wasserstein* (SSFG), that can find an important area of projections characterized by a von Mises-Fisher distribution. Then, we introduce two variants of SSFG to improve its performance. The first variant, named *mixture spherical sliced fused Gromov Wasserstein* (MSSFG), replaces the vMF distribution by a mixture of von Mises-Fisher distributions to capture multiple important areas of directions that are far from each other. The second variant, named *power spherical sliced fused Gromov Wasserstein* (PSSFG), replaces the vMF distribution by a power spherical distribution to improve the sampling time in high dimension settings. We then apply the new discrepancies to the RAE framework to achieve its new variants. Finally, we conduct extensive experiments to show that the new proposed autoencoders have favorable performance in learning latent manifold structure, image generation, and reconstruction.

## 1 INTRODUCTION

In recent years, autoencoders have been used widely as important frameworks in several machine learning and deep learning models, such as generative models (Kingma & Welling, 2013; Tolstikhin et al., 2018; Kolouri et al., 2018) and representation learning models (Tschannen et al., 2018). Formally, autoencoders consist of two components, namely, an encoder and a decoder. The encoder denoted by $E_\phi$ maps the data, which is presumably in a low dimensional manifold, to a latent space. Then the data could be generated by sampling points from the latent space via a prior distribution $p$, then decoding those points by the decoder $G_\theta$. The decoder is formally a function from latent space to the data space and it induces a distribution $p_{G_\theta}$ on the data space. In generative modeling, the major task is to obtain a decoder $G_{\theta^*}$ such that its induced distribution $p_{G_{\theta^*}}$ and the data distribution are very close under some discrepancies. Two popular instances of autoencoders are the variational autoencoder (VAE) (Kingma & Welling, 2013), which uses KL divergence, and the Wasserstein autoencoder (WAE) (Tolstikhin et al., 2018), which chooses the Wasserstein distance (Villani, 2008) as the discrepancy between the induced distribution and the data distribution.

---

[*]The work was finished when Nhat Ho worked at VinAI Research in the summer of 2020.

In order to implement the WAE, a relaxed version was introduced by removing the constraint on the prior and the aggregated posterior (latent code distribution). In particular, a chosen discrepancy between these distributions is added to the objective function and plays a role as a regularization term. With that relaxation approach, the WAE becomes a flexible framework for customized choices of the discrepancies (Patrini et al., 2020; Kolouri et al., 2018). However, the WAE suffers either from the over-regularization problem when the prior distribution is too simple (Dai & Wipf, 2018; Ghosh et al., 2019), which is usually chosen to be isotropic Gaussian, or from the under-regularization problem when learning an expressive prior distribution jointly with the autoencoder without additional regularization, e.g., structural regularization (Xu et al., 2020). In order to circumvent these issues of WAE, relational regularized autoencoder (RAE) was proposed in (Xu et al., 2020) with two major changes. The first change is to use a mixture of Gaussian distributions as the prior while the second change is to set a regularization on the structural difference between the prior and the aggregated posterior distribution, which is called the relational regularization. The state-of-the-art version of RAE, deterministic relational regularized autoencoder (DRAE), utilizes the sliced fused Gromov Wasserstein (SFG) (Xu et al., 2020) as the relational regularization. Although DRAE performs well in practice and has good computational complexity (Xu et al., 2020), the SFG does not fully exploit the benefits of relational regularization due to its slicing drawbacks. Similar to sliced Wasserstein (SW) (Bonnotte, 2013; Bonneel et al., 2015) and sliced Gromov Wasserstein (SG) (Vayer et al., 2019), SFG uses the uniform distribution over the unit sphere to sample projecting directions. However, that leads to the underestimation of the discrepancy between two target distributions (Deshpande et al., 2019; Kolouri et al., 2019) since many unimportant directions are included in that estimation. A potential solution is by using only the best Dirac measure over the unit sphere to sample projecting directions in SFG, which was employed in max-sliced Wasserstein distance Deshpande et al. (2019). However, this approach focuses on the discrepancy of the target probability measures based on only one important direction while other important directions are not considered. As one alternative solution, authors in (Nguyen et al., 2021) proposed the distributional slicing approach which is a general technique to design a probabilistic way to select important directions.

**Our contributions.** To improve the effectiveness of the relational regularization in the autoencoder framework, we propose novel sliced relational discrepancies between the prior and the aggregated posterior. The new sliced discrepancies utilize von Mises-Fisher distribution and its variants instead of the uniform distribution as the distributions over slices. An advantage of the vMF distribution and its variants is that they could interpolate between the Dirac measure and uniform measure, thereby improving the quality of the projections sampled from these measures and overcoming the weaknesses of both the SFG and its max version—max-SFG. In summary, our major contributions are as follows:

1. First, we propose a novel discrepancy, named *spherical sliced fused Gromov Wassersetein* (SSFG). This discrepancy utilizes vMF distribution as the slicing distribution to focus on the area of directions that can separate the target probability measures on the projected space. Moreover, we show that SSFG is a well-defined pseudo-metric on the probability space and does not suffer from the curse of dimensionality for the inference purpose. With favorable theoretical properties of SSFG, we apply it to the RAE framework and obtain a variant of RAE, named *spherical deterministic RAE* (s-DRAE).

2. Second, we propose an extension of SSFG to *mixture SSFG* (MSSFG) where we utilize a mixture of vMF distributions as the slicing distribution (see Appendix C for the details). Comparing to the SSFG, the MSSFG is able to simultaneously search for multiple areas of important directions, thereby capturing more important directions that could be far from each other. Based on the MSSFG, we then propose another variant of RAE, named *mixture spherical deterministic RAE* (ms-DRAE).

3. Third, to improve the sampling time and stability of vMF distribution in high dimension settings, we introduce another variant of SSFG, named *power SSFG* (PSSFG), which uses power spherical distribution instead of the vMF distribution as the slicing distribution. Then, we apply the PSSFG to the RAE framework to obtain the *power spherical deterministic RAE* (ps-DRAE).

4. Finally, we carry out extensive experiments on standard datasets to show that proposed autoencoders achieve the best generative quality among well-known autoencoders, including the state-of-the-art RAE—DRAE. Furthermore, the experiments indicate that the s-DRAE, ms-DRAE, and ps-DRAE can learn a nice latent manifold structure, a good mixture of

Gaussian prior which can cover well the latent manifold, and provide more stable results in both generation and reconstruction than DRAE.

We note in passing that the proposed sliced-fused Gromov-Wasserstein divergences are not just applicable to the applications within the RAE framework. In Appendix E.6, we provide initial experiments to show that these divergences can be used to improve over other sliced-based distances in color transfer applications. We leave a thorough extension of the proposed sliced-fused discrepancies to other applications in the future work.

**Organization.** The remainder of the paper is organized as follows. In Section 2, we provide backgrounds for DRAE and vMF distribution. In Section 3, we propose the spherical sliced fused Gromov Wasserstein and its extension. We then apply these spherical discrepancies to the relational regularized autoencoder. Extensive experiment results are presented in Section 4 followed by conclusion in Section 5. Proofs of key results and extra materials are in the supplementary material.

**Notation:** Let $\mathbb{S}^{d-1}$ be the $d$-dimensional hypersphere and $\mathcal{U}(\mathbb{S}^{d-1})$ be the uniform distribution on $\mathbb{S}^{d-1}$. For a metric space $(\mathcal{X}, d_1)$, we denote by $\mathcal{P}(\mathcal{X})$ the space of probability distributions over $\mathcal{X}$ with finite moments. We say that $d_1$ is a pseudo-metric in space $\mathcal{X}$ if it is non-negative, symmetric, and satisfies the inequality: $d_1(x, z) \leq C[d_1(x, y) + d_1(y, z)]$ for a universal constant $C > 0$ and for all $x, y, z \in \mathcal{X}$. For any distribution $\mu$ and $\nu$, $\Pi(\mu, \nu)$ is the set of all transport plans between $\mu$ and $\nu$. For $x \in \mathbb{R}^d$, denote $\delta_x$ to be the Dirac measure at $x$. For any $\theta \in \mathbb{S}^{d-1}$ and any measure $\mu$, $\theta \sharp \mu$ denotes the pushforward measure of $\mu$ through the mapping $\mathcal{R}_\theta$ where $\mathcal{R}_\theta(x) = \theta^\top x$ for all $x$.

## 2 BACKGROUND

In this section, we provide backgrounds for the sliced fused Gromov Wasserstein and the relational regularized autoencoders. Then, we give backgrounds for the von Mises-Fisher distribution.

### 2.1 DETERMINISTIC RELATIONAL REGULARIZED AUTOENCODER AND SLICED FUSED GROMOV WASSERSTEIN

First, we review the WAE framework (Tolstikhin et al., 2018), which is used to learn a generative model by minimizing a relaxed version of Wasserstein distance (Villani, 2008) between data distribution $p_d(x)$ and model distribution $p_\theta(x) := G_\theta \sharp p(z)$, where $p(z)$ is a noise distribution. The model aims to find the autoencoder which solves the following objective function:

$$\min_{\theta, \phi} \mathbb{E}_{p_d(x)} \mathbb{E}_{q_\phi(z|x)}[d(x, G_\theta(z))] + \lambda D(q_\phi(z)||p(z)), \tag{1}$$

where $d$ is the ground metric of Wasserstein distance, $D$ is a discrepancy between distributions, and $q_\phi(z|x)$ is a distribution for encoder $E_\phi : X \to Z$, parameterized by $\phi$. Due to the efficiency in training generative models, several autoencoder models are derived from this framework. For example, WAE uses standard Gaussian distribution for $p(z)$ and chooses $D$ to be either maximum mean discrepancy (MMD) or GAN. Later, by using sliced Wasserstein distance (Bonneel et al., 2015) for $D$, Kolouri et al. (2018) achieved another type of autoencoder, which is called SWAE.

**Deterministic relational regularized autoencoder (DRAE):** In DRAE, Xu et al. (2020) parametrizes the prior as a mixture of Gaussians $(p_{\mu_{1:k}, \Sigma_{1:k}}(z))$ and makes it learnable. Additionally, they introduce the sliced fused Gromov Wasserstein as the discrepancy between the posterior and the prior distributions.

**Definition 1.** *(SFG) Let $\mu, \nu \in \mathcal{P}(\mathbb{R}^d)$ be two probability distributions, $\beta$ be a constant in $[0, 1]$, and $d_1 : \mathbb{R} \times \mathbb{R} \to \mathbb{R}_+$ be a pseudo-metric on $\mathbb{R}$. The **sliced fused Gromov Wasserstein** (SFG) between $\mu$ and $\nu$ is defined as:*

$$SFG(\mu, \nu; \beta) := \mathbb{E}_{\theta \sim \mathcal{U}(\mathbb{S}^{d-1})}[D_{fgw}(\theta \sharp \mu, \theta \sharp \nu; \beta, d_1)], \tag{2}$$

*where the **fused Gromov Wasserstein** $D_{fgw}$ is given by:*

$$D_{fgw}(\theta \sharp \mu, \theta \sharp \nu; \beta, d_1) := \min_{\pi \in \Pi(\theta \sharp \mu, \theta \sharp \nu)} \left\{ (1-\beta) \int_{\mathbb{R}^d \times \mathbb{R}^d} d_1(\theta^\top x, \theta^\top y) d\pi(x, y) \right.$$
$$\left. + \beta \int_{(\mathbb{R}^d)^4} \left[ d_1(\theta^\top x, \theta^\top x') - d_1(\theta^\top y, \theta^\top y') \right]^2 d\pi(x, y) d\pi(x', y') \right\}. \tag{3}$$

Given the definition of SFG, the objective function of the deterministic relational regularized autoencoder (DRAE) takes the following form:

$$\min_{\theta, \phi, \mu_{1:k}, \Sigma_{1:k}} \mathbb{E}_{p_d(x)} \mathbb{E}_{q_\phi(z|x)} \big[ d\big(x, G_\theta(z)\big) \big] + \lambda \mathbb{E}_{q_\phi(z), p_{\mu_{1:k}, \Sigma_{1:k}}(z)} SFG\big[\big(\hat{q}_N(z) || \hat{p}_N(z)\big)\big], \quad (4)$$

where $\hat{q}_N(z)$ and $\hat{p}_N(z)$ are the empirical distributions of $q_\phi(z)$ and $p_{\mu_{1:k}, \Sigma_{1:k}}(z)$ respectively.

**Properties of SFG:** From equation (2), SFG is a linear combination of sliced Wasserstein (SW) and sliced Gromov Wasserstein (SG). In particular, SFG becomes SW and SG when $\beta = 0$ and $\beta = 1$, respectively. Hence SFG is able to take advantages of both of them. If $\mu$ and $\nu$ have $n$ supports and uniform weights and $d_1(x, y) = (x - y)^2$, SFG has computational complexity of the order $\mathcal{O}(n \log n)$. It is because under $d_1$, both SW and SG have closed-form expressions (Vayer et al., 2019; Bonnotte, 2013) where the optimal transport map $\pi$ in $D_{fgw}$ (Vayer et al., 2018) can be obtained by sorting the projected supports of $\mu$ and $\nu$.

**Limitation of SFG:** The major limitation of SFG is that the outer expectation with respect to $\theta \sim \mathcal{U}(\mathbb{S}^{d-1})$ in SFG is generally intractable. In practice, projections from the unit sphere are uniformly sampled and we then apply the Monte Carlo method to obtain an approximate of that expectation. However, the difference between two distributions is certainly not distributed uniformly, meaning that informative directions are mixed up with many non-informative ones. Hence, sampling blindly slices in high dimensional space not only is ineffective but also underestimates the discrepancy between the two distributions. The von Mises-Fisher (vMF) distribution provides a way to have concentrated weight on the most important directions and assigns less weight to further directions. Therefore, we gain a better representation of the discrepancy between probability measures.

## 2.2 VON MISES-FISHER DISTRIBUTION

Now, we review the definition of the von Mises-Fisher distribution.

**Definition 2.** *The von Mises–Fisher distribution (vMF) is a probability distribution on the unit sphere $\mathbb{S}^{d-1}$ where its density function is given by (Jupp et al., 1979):*

$$f(x|\epsilon, \kappa) := C_d(\kappa) \exp(\kappa \epsilon^\top x), \quad (5)$$

*where $\kappa \geq 0$ is the concentration parameter, $\epsilon \in \mathbb{S}^{d-1}$ is the location vector, and $C_d(\kappa) := \frac{\kappa^{d/2-1}}{(2\pi)^{d/2} I_{d/2-1}(\kappa)}$ is the normalization constant. Here, $I_{d/2-1}$ is the modified Bessel function of the first kind at order $d/2 - 1$ (Temme, 2011).*

The vMF concentrates around mode $\epsilon$ and its density decreases when $x$ goes away from $\epsilon$. When $\kappa \to 0$, vMF converges to the uniform distribution, and when $\kappa \to \infty$, vMF approaches to the Dirac distribution centered at $\epsilon$ (Sra, 2016). These properties are illustrated by a toy example in Figure 3 in Appendix B.1.

## 3 SPHERICAL SLICED FUSED GROMOV WASSERSTEIN AND ITS RELATIONAL REGULARIZED AUTOENCODER

In this section, we introduce a novel discrepancy, named *spherical sliced fused Gromov Wasserstein* (SSFG), that searches for the best vMF distribution which distributes more masses to the most important area of projections on the unit sphere $\mathbb{S}^{d-1}$. Then, we discuss an application of SSFG to the relational regularized autoencoder framework.

### 3.1 SPHERICAL SLICED FUSED GROMOV WASSERSTEIN

We first start with a definition of spherical sliced fused Gromov Wasserstein.

**Definition 3.** *(SSFG) Let $\mu, \nu \in \mathcal{P}(\mathbb{R}^d)$ be two probability distributions, $\kappa > 0$, $\beta \in [0, 1]$, $d_1 : \mathbb{R} \times \mathbb{R} \to \mathbb{R}_+$ be a pseudo-metric on $\mathbb{R}$. The **spherical sliced fused Gromov Wasserstein (SSFG)** between $\mu$ and $\nu$ is defined as follows:*

$$SSFG(\mu, \nu; \beta, \kappa) := \max_{\epsilon \in \mathbb{S}^{d-1}} \mathbb{E}_{\theta \sim \text{vMF}(\cdot|\epsilon, \kappa)} \big[ D_{fgw}(\theta \sharp \mu, \theta \sharp \nu; \beta, d_1) \big], \quad (6)$$

*where the fused Gromov Wasserstein $D_{fgw}$ is defined at equation (3).*

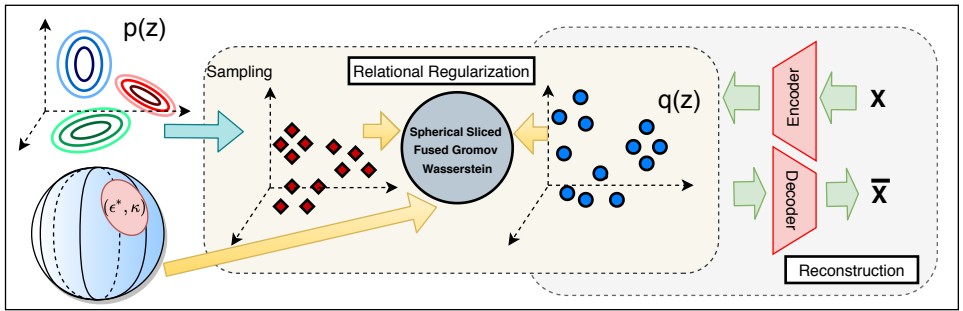

Figure 1: Visualization of the spherical deterministic relational regularized autoencoder (s-DRAE).

A few comments on the SSFG are in order. The family of vMF distributions is controlled by two parameters $\epsilon$ and $\kappa$, where $\epsilon$ is the mode of the vMF distribution while $\kappa$ controls its concentration. By changing $\kappa$ from 0 to infinity, the vMF family could interpolate from the uniform distribution to any Dirac distribution on the sphere. In other words, it allows us to control distributing weight to the most important direction and other directions based on the geodesic distance on the sphere. Optimizing over the family of vMF distributions helps us to identify where the best direction is as well as how much weight we need to put there in comparison with other less important directions.

**Sampling procedure and reparameterization trick with the vMF:** To generate samples from vMF, we follow the procedure in (Ulrich, 1984), which is described in Algorithm 1 in Appendix B. Note that this procedure does not suffer from the curse of dimensionality. Furthermore, to compute integral with respect to the vMF distribution, we use the reparametrization scheme in (Naesseth et al., 2017), which was extended for vMF in Lemma 2 in (Davidson et al., 2018). Finally, Davidson et al. (2018) proved that samples from Algorithm 1 (cf. Appendix B.2) can provide a differentiable estimator for the parameters of vMF distribution. More details of this scheme are given in Appendix B.3.

**Complexity of computing SSFG:** Let $\mu$ and $\nu$ be two discrete distributions that have $n$ supports with uniform weights. For the general case of $d_1$, similar to SFG, the complexity of computing SSFG can be expensive (at least of the order $\mathcal{O}(n^4)$ as the fused Gromov Wasserstein $D_{fgw}$ is a quadratic programming problem). However, the complexity of SSFG can be greatly improved under specific choices of $d_1$. For example, when $d_1(x, y) = (x - y)^2$, with a similar argument to the SFG case, the SSFG has computational complexity of the order $\mathcal{O}(n \log n)$.

**Key properties of SSFG:** We first prove that SSFG is a pseudo-metric on the probability space.

**Theorem 1.** *For any $\beta \in [0, 1]$ and $\kappa > 0$, $SSFG(., .; \beta, \kappa)$ is a pseudo-metric in the space of probability measures, namely, it is non-negative, symmetric, and satisfies the weak triangle inequality.*

The proof of Theorem 1 is in Appendix A.1. Our next result establishes relations between SSFG and SFG and the max version of SFG, named as max-SFG.

**Theorem 2.** *For any probability measures $\mu, \nu \in \mathcal{P}(\mathbb{R}^d)$, the following holds:*

*(a)*
$$\lim_{\kappa \to 0} SSFG(\mu, \nu; \beta, \kappa) = SFG(\mu, \nu; \beta),$$
$$\lim_{\kappa \to \infty} SSFG(\mu, \nu; \beta, \kappa) = \max_{\theta \in \mathbb{S}^{d-1}} D_{fgw}(\theta\sharp\mu, \theta\sharp\nu; \beta) := \textit{max-SFG}(\mu, \nu; \beta).$$

*(b) For any $\kappa > 0$, we find that*

$$\exp(-\kappa)C_d(\kappa)SFG(\mu, \nu; \beta) \leq SSFG(\mu, \nu; \beta, \kappa) \leq \exp(\kappa)C_d(\kappa)SFG(\mu, \nu; \beta),$$
$$SSFG(\mu, \nu; \beta, \kappa) \leq \textit{max-SFG}(\mu, \nu; \beta).$$

The proof of Theorem 2 is in Appendix A.2. Theorem 2 shows that SSFG is an interpolation between SFG and max-SFG, namely, it combines the properties of both SFG and max-SFG. Furthermore, the result of part (b) of Theorem 2 indicates that SSFG is strongly equivalent to SFG.

Our next result shows that SSFG does not suffer from the curse of dimensionality for the inference purpose under certain choices of $d_1$. Therefore, it will be a statistically efficient discrepancy to compare the prior distribution to the encoder distribution in the DRAE framework.

**Theorem 3.** *Assume that $\mu$ is a probability measure supported on a compact subset $\Theta \subset \mathbb{R}^d$. Let $X_1, \ldots, X_n$ be i.i.d. data from $P$ and $d_1(x, y) = |x - y|^r$ for a positive integer $r$. We denote by $\mu_n = \frac{1}{n} \sum_{i=1}^n \delta_{X_i}$ the empirical measure of the data points $X_1, \ldots, X_n$. Then, for any $\beta \in [0, 1]$ and $\kappa > 0$, there exists a constant $c$ depending only on $r$ and diameter of $\Theta$ such that*

$$\mathbb{E}\Big[SSFG(\mu_n, \mu; \beta, \kappa)\Big] \leq \frac{c}{n}.$$

Theorem 3 together with the earlier argument about the computational complexity of SSFG suggests that the choice of $d_1(x, y) = (x - y)^2$ is not only convenient for the computation but also statistically efficient. Therefore, we will specifically use this choice of $d_1$ in our experiments in Section 4.

**Spherical deterministic relational regularized autoencoder:** We replace SFG by SSFG in the deterministic relational regularized autoencoder framework in equation (4) to obtain a new variant of DRAE with a stronger relational regularization. The new autoencoder is named as *spherical deterministic relational regularized autoencoder* (s-DRAE). Intuitive visualization of s-DRAE is presented in Figure 1. The detailed training procedure for s-DRAE is left in Appendix B.5.

## 3.2   EXTENSIONS AND VARIANTS OF SSFG

We first propose an extension of SSFG to its mixture variant.

**Definition 4.** *(MSSFG) Let $\mu, \nu \in \mathcal{P}(\mathbb{R}^d)$ be two probability distributions, $\beta \in [0, 1]$ be a constant, $\{\alpha_i\}_{i=1}^k$ be given mixture weights, and $\{\kappa_i\}_{i=1}^k$ be given mixture concentration parameters where $k \geq 1$. Furthermore, let $d_1 : \mathbb{R} \times \mathbb{R} \to \mathbb{R}_+$ be a pseudo-metric on $\mathbb{R}$. Then, the **mixture spherical sliced fused Gromov Wasserstein** (MSSFG) between $\mu$ and $\nu$ is defined as follows:*

$$MSSFG(\mu, \nu; \beta, \{\kappa_i\}_{i=1}^k, \{\alpha_i\}_{i=1}^k)$$
$$:= \max_{\epsilon_{1:k} \in \mathbb{S}^{d-1}} \mathbb{E}_{\theta \sim MovMF(\cdot|\epsilon_{1:k}, \{\kappa_i\}_{i=1}^k, \{\alpha_i\}_{i=1}^k)} \Big[D_{fgw}(\theta \sharp \mu, \theta \sharp \nu; \beta, d_1)\Big], \quad (7)$$

*where $D_{fgw}$ is defined in equation (3) and the mixture of vMF distributions is defined as $MovMF(\cdot|\epsilon_{1:k}, \{\kappa_i\}_{i=1}^k, \{\alpha_i\}_{i=1}^k) := \sum_{i=1}^k \alpha_i vMF(\cdot|\epsilon_i, \kappa_i)$.*

**Comparison between MSSFG and SSFG:**   When $k = 1$, the MSSFG becomes SSFG. Recall that SSFG tries to search for the best location parameter in the unit sphere $\mathbb{S}^{d-1}$ that maximizes the expected value of the fused Gromov Wasserstein between the projected probability measures. Intuitively, it places a large weight on the best projection and some weights on other important projections. However, if these important projections are far from the best projection, i.e., the center of the best von Mises-Fisher distribution, their weights will be very small, which can be undesirable. To account for this issue, the mixture of von Mises-Fisher distributions aims to find $k$ best location parameters whose weights are guaranteed to be large enough. Furthermore, when $k$ is chosen to be sufficiently large, mixture of von Mises-Fisher distributions will give a good coverage of the unit sphere; therefore, the important directions that MSSFG can find will be able to reflect more accurate differences between the target probability distributions than those from SSFG.

**Properties of MSSFG and its DRAE version:** As SSFG, MSSFG is a pseudo-metric in the probability space and does not suffer from the curse of dimensionality. Its computational complexity is of the order $\mathcal{O}(n \log n)$ when $\mu$ and $\nu$ are discrete measures with $n$ atoms and uniform weights and $d_1(x, y) = (x - y)^2$. The detailed discussion of the properties of MSSFG is in Appendix C. An application of MSSFG to the DRAE framework leads to the *mixture spherical DRAE* (ms-DRAE).

**Improving computational time of (M)SSFG:** Drawing the samples from the vMF distribution and its mixtures can be slow in high dimension settings, which affects the computation of (M)SSFG. To account for this issue, we propose using *power spherical distribution* (De Cao & Aziz, 2020) instead of vMF and its mixtures as the slicing distribution to improve the computational time of (M)SSFG. It leads to a new discrepancy, named *power SSFG* (PSSFG), between the probability distributions (see Appendix D for the definition). In Section 4, we show that (M)PSSFG has better computational time than (M)SSFG while its DRAE version, named *(mixture) power spherical DRAE* ((m)ps-DRAE), has comparable performance to (m)s-DRAE.

Table 1: FID scores and reconstruction losses of different autoencoders. (*) denotes the results that are taken from (Xu et al., 2020) due to the reproducing failure. The results are taken from 5 different runs.

| Method | MNIST | | CelebA | |
|---|---|---|---|---|
| | FID | Reconstruction | FID | Reconstruction |
| VAE | $71.55 \pm 26.65$ | $18.59 \pm 2.22$ | $59.99(*)$ | $96.36(*)$ |
| GMVAE | $75.68 \pm 11.95$ | $18.19 \pm 0.14$ | $212.59 \pm 18.15$ | $97.77 \pm 0.19$ |
| Vampprior | $138.03 \pm 34.09$ | $29.98 \pm 4.09$ | - | - |
| PRAE | $100.25 \pm 41.72$ | $16.20 \pm 3.14$ | $52.20~(*)$ | $\mathbf{63.21(*)}$ |
| WAE | $80.77 \pm 11$ | $11.53 \pm 0.33$ | $52.07~(*)$ | $63.83(*)$ |
| SWAE | $80.28 \pm 19.22$ | $14.12 \pm 2.06$ | $86.53 \pm 2.49$ | $89.71 \pm 2.15$ |
| DRAE | $58.04 \pm 20.74$ | $14.07 \pm 4.31$ | $50.09 \pm 1.33$ | $66.05 \pm 2.56$ |
| m-DRAE (ours) | $52.92 \pm 13.81$ | $13.13 \pm 0.33$ | $49.05 \pm 0.93$ | $66.30 \pm 0.22$ |
| s-DRAE (ours) | $47.97 \pm 13.83$ | $11.17 \pm 1.73$ | $46.63 \pm 0.83$ | $66.62 \pm 0.51$ |
| ps-DRAE (ours) | $49.15 \pm 12.93$ | $11.71 \pm 1.21$ | $48.21 \pm 1.02$ | $66.31 \pm 0.43$ |
| mps-DRAE (ours) | $44.67 \pm 9.98$ | $11.01 \pm 1.32$ | $46.61 \pm 1.01$ | $66.23 \pm 0.56$ |
| ms-DRAE (ours) | $\mathbf{43.57 \pm 10.98}$ | $\mathbf{11.12 \pm 0.91}$ | $\mathbf{46.01 \pm 0.91}$ | $65.91 \pm 0.4$ |

## 4 EXPERIMENTS

In this section, we conduct extensive experiments on MNIST (LeCun et al., 1998) and CelebA datasets (Liu et al., 2015) to evaluate the performance of s-DRAE, ps-DRAE and m(p)s-DRAE with various autoencoders, including DRAE (trained by SFG), PRAE (Xu et al., 2020), m-DRAE (trained by max-SFG—see its definition in Theorem 2), VAE (Kingma & Welling, 2013), WAE (Tolstikhin et al., 2018), SWAE (Kolouri et al., 2018), GMVAE (Dilokthanakul et al., 2016), and the VampPrior (Tomczak & Welling, 2018). We use two standard scores as evaluation metrics: (i) the Frechet Inception distance (FID) score (Heusel et al., 2017) is used to measure the generative ability; (ii) the reconstruction score is used to evaluate the reconstruction performance computed on the test set. For the computational details of the FID score, we compute the score between 10000 randomly generated samples and all samples from the test set of each dataset. To guarantee the fairness of the comparison, we use the same autoencoder architecture, Adam optimizer with learning rate = 0.001, $\beta_1 = 0.5$ and $\beta_2 = 0.999$; batch size = 100; latent size = 8 on MNIST and 64 on CelebA; coefficient $\lambda$=1; fused parameter $\beta = 0.1$. We set the number of components $K = 10$ for autoencoder with a mixture of Gaussian distribution as the prior. More detailed descriptions of these settings are in Appendix F. **Comparing with other autoencoders:** We first report the performances of autoencoders on MNIST (LeCun et al., 1998) and CelebA datasets (Liu et al., 2015). Table 1 presents the FID scores and reconstruction losses of trained autoencoders. All results are obtained from five different runs and reported with empirical mean and standard deviation. On the MNIST dataset, ms-DRAE achieves the lowest scores in both FID score and reconstruction loss among all the autoencoders. In addition, s-DRAE and (m)ps-DRAE also have better scores than DRAE. On the CelebA dataset, we cannot reproduce results from VAE, PRAE, and WAE; therefore, we use the results with these autoencoders from DRAE paper (Xu et al., 2020). Table 1 suggests that ms-DRAE also obtains the lowest mean and standard deviation in FID score than other autoencoders, meanwhile its reconstruction loss is almost the same as other DRAEs. The FID scores of s-DRAE and ps-DRAE are also better than those of DRAE. These results suggest that the proposed spherical discrepancies truly improve the performances of the DRAE framework. In these experiments, we set the number of projections $L = 50$ for every sliced-discrepancy. For s-DRAE, ps-DRAE and m(p)s-DRAE (10 vMF components with uniform weights and same concentration parameters), we search for $\kappa \in \{1, 5, 10, 50, 100\}$ which gives the best FID score on the validation set of the corresponding dataset. By tuning $\kappa$, we find that the performance of both s-DRAE and ps-DRAE is close to that of DRAE when $\kappa \in \{0.001, 0.01, 0.1, 1\}$, namely, the reconstruction loss and FID score are nearly equal to the scores of DRAE. On the other extreme, when $\kappa = 100$, s-DRAE and ps-DRAE behave like m-DRAE in both evaluation metrics. Further details are given in Figures 12 and 15 in Appendices E.1 and E.3 respectively.

Detailed results including generated images, reconstruction images and visualizing latent spaces are in Appendix E.1. These results indicate that s-DRAE, ps-DRAE, and ms-DRAE can learn nice latent

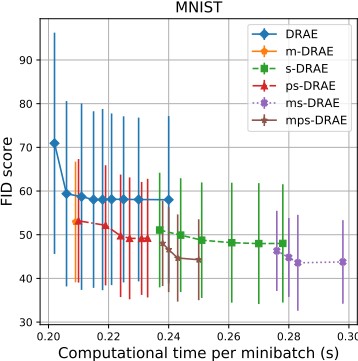 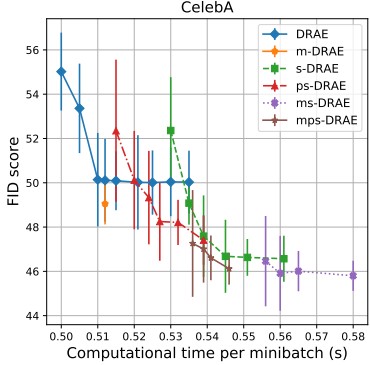

Figure 2: Each dot represents the computational time per minibatch and FID score. For DRAE, we vary the number of projections $L \in \{1, 5, 10, 20, 50, 100, 200, 500, 1000\}$; for s-DRAE we set $\kappa = 10$, $L \in \{1, 5, 10, 20, 50, 100\}$; for ps-DRAE we set $\kappa = 50, L \in \{1, 5, 10, 20, 50, 100\}$; and for m(p)s-DRAE we set $L = 50$, the number of vMF components $k \in \{2, 5, 10, 50\}$ (for each $k$, we find the best $\kappa \in \{1, 5, 10, 50, 100\}$).

structures and mixture Gaussian priors which can cover well these latent spaces. As a consequence, the spherical DRAEs can produce good generated images and reconstruct images correctly.

**Detailed comparisons among deterministic RAEs:** It is well-known that the quality and computational time of sliced-discrepancies depend on the number of projections (Kolouri et al., 2018; 2019; Deshpande et al., 2019). Therefore, we carry out experiments on MNIST and CelebA datasets to compare ps-DRAE, m(p)s-DRAE, s-DRAE to DRAE in a wide range the of number of projections. In detail, we set the number of projections $L \in \{1, 5, 10, 20, 50, 100, 200, 500, 1000\}$ in DRAE ; and $L \in \{1, 5, 10, 20, 50, 100\}$ for s-DRAE ($\kappa = 10$), and ps-DRAE ($\kappa = 50$). We then report the (minibatch) computation time and FID score of these autoencoders in Figure 2. In this figure, we also plot the time and FID scores of m-DRAE and m(p)s-DRAE (using $L = 50$ and the number of vMF components $k \in \{2, 5, 10, 50\}$. On the MNIST dataset, with only 1 projection, s-DRAE achieves lower FID score than all settings of DRAE; however, s-DRAE requires more time to train due to the sampling of vMF and its optimization problem. Having faster sampling procedure of PS distribution, ps-DRAE has better computational time than s-DRAE while it still has a comparable performance to s-DRAE. With the computational time greater than about 0.21(s), ps-DRAE always produces lower FID score than DRAE. We also observe the same phenomenon on CelebA dataset, namely, ps-DRAE and s-DRAE have lower FID score than DRAE with any value of $L$ but $L = 1$. Between s-DRAE and ps-DRAE, s-DRAE gives better results but ps-DRAE is faster in training. On both the datasets, m-DRAE has faster speed than (p)s-DRAE but its FID score is higher. In terms of the FID score, ms-DRAE is the best autoencoder though it can be more expensive in training. Moreover, mps-DRAE has more efficient computational time than ms-DRAE while its FID scores are comparable. Finally, we observe that increasing the number of components in m(p)s-DRAE can enhance the FID score but also worsen the computational speed.

**Additional experiments:** We provide further comparisons between MSSFG, PSSFG, SSFG and SFG in ex-post density estimation of autoencoder (Ghosh et al., 2019) and GAN (Goodfellow et al., 2014) applications in Appendices E.4 and E.5. In the ex-post density estimation framework, we find that MSSFG, PSSFG and SSFG give better FID score than SFG. Like traditional training procedures, MSSFG achieves the best performance in this task. In the GAN application, we use a toy example, which is to learn a generator to produce 4 Gaussian modes. We observe that MSSFG, PSSFG and SSFG help the model distribution converge faster than SFG does. Finally, we also apply the proposed sliced-discrepancies to image color adaption (Rabin et al., 2014; Bonneel et al., 2015; Perrot et al., 2016) in Appendix E.6, where we find that using (M)vMF, (M)PS distributions to sample projecting directions can improve the performance of the sliced-based color adaption algorithms (Rabin et al., 2010; Bonneel et al., 2015; Muzellec & Cuturi, 2019).

## 5 CONCLUSION

In the paper, we first introduced a new spherical relational discrepancy, named spherical sliced fused Gromov Wasserstein (SFFG), between the probability measures. This discrepancy is obtained by

replacing the uniform distribution over slicing direction in sliced fused Gromov Wasserstein by a von Mises-Fisher distribution that can cover the most informative area of directions. To improve the performance and stability of SFFG, we then propose two variants of SSFG: (i) the first variant is mixture SSFG (MSSFG), obtained by using a mixture of vMF distributions instead of a single vMF distribution to capture more informative areas of directions; (ii) the second variant is power SSFG (PSSFG), obtained by replacing the vMF distribution by the power spherical distribution to improve the sampling time of vMF distribution in high dimension settings. An application of these discrepancies to the DRAE framework leads to several new variants of DRAE. Extensive experiments show that these new autoencoders are more stable and achieve better generative performance than the previous autoencoders, including DRAE, in comparable computational time.

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
