# OpenReview forum: "Improving Relational Regularized Autoencoders with Spherical Sliced Fused Gromov Wasserstein"
_ICLR.cc/2021/Conference — ICLR 2021 Poster_

### Official Review · AnonReviewer1 · 2020-10-28
**Original work but maybe a little bit incremental**

**Rating:** 7
**Confidence:** 5

**Review:**

This paper builds on the work of Xu and colleagues (2020) on auto encoders (AE) with relational regularization. The core idea is to enforce a notion of structure in the latent space of an AE, by measuring the (composed, structrural) divergence with a target distribution. In order to do so, a quadratic optimal transport problem is used (the fused Gromov-Wasserstein -FGW-), which has a super-cubical complexity. In order to alleviate this cost, Xu et al. defined a sliced version of it (akin to sliced Wasserstein -SW-), which consists in solving several simple 1D versions of the problem, that admit a close form solution, after projecting onto random directions drawn uniformly on the unit hypersphere. This work proposes to replace this uniform distribution over the sphere by a von Mises-Fisher distribution on the sphere, that is alike a ‘Gaussian distribution’ on the sphere, and also a mixture of those distributions. Their parameters are optimized during the training so that it maximises the  FGW divergence. This strategy is similar to recent trends in computing SW, that also replace the uniform distribution by either the replacing the expectation by a max or looking for subspaces that maximizes this the expected SW.

The paper is clearly written and interesting. It can be seen as incremental with respect to the work of Xu et al., but the formulation of there sliced optimal transport problem with a parametrized von Mises-Fisher distribution is novel and could also been applied to the original sliced Wasserstein, but maybe also to compute the sliced Fused Gromov Wasserstein for other task such as graph classification, as in the original FGW paper (see minor comment).
The experimental results are very good and clearly show the benefits of the method. Yet I have some questions, which answers might be critical for the final evaluation of the paper :
 - in the mixture part, how do you train for the parameters of the mixture ? Do you have a kind of EM algorithm, or do you perform a gradient ascent ?
 - in general, directions are drawn randomly for every batches of samples. What is the meaning of fixing in advance the number of projections, as done in Figure 3 and the related experiment ?
 - it seems that the sliced FGW is computed on mini-batches of samples. While I acknowledge there is a common practice to do so, a 1D FGW on a mini batch is not the same as computing the 1D FGW on the full dataset. As such, this mini batch version of FGW is not the same as computing the true FGW. In the end, i) the size of the mini-batch might have an impact on the estimation quality, that should be discussed ii) if computing 1D FGW on mini batches, why not computing and comparing with the mini batch version of the original version of FGW ? From the paper, the batch size is 100, this would be solved very quickly by current FGW solvers.

In the end, I think that using the von Mises Fisher distribution is an interesting and original idea, which might have a broader impact than the sliced FGW. The paper has also the merit to push a little forward the structural regularized AE which is, in the reviewer’s opinion, a good point. On the negative side, this work can be considered a little bit as incremental, and some questions remain on the experimental part.  I am willing to change my rating depending on the answers to my comments.

Minor remarks

It is interesting to note that another paper in consideration for ICLR is developing a similar strategy in the case of the simple sliced Wasserstein distance:
https://openreview.net/forum?id=QYjO70ACDK
It is also a little bit strange that the original fused Gromov-Wasserstein paper is not credited in the paper:
[1] Titouan, Vayer, et al. "Optimal Transport for structured data with application on graphs." International Conference on Machine Learning. 2019.

### After author response
I thank the authors for their detailed response to my comments. I do not agree with the complexity of FGW being solved in $n^4$ which should be more related to $n^3$ for the type of distance considered in the paper (see analysis in [1]), but yet the point is still sensible for considering the minibatch version. My other comments have been adressed, and I am changing my note to the score of 7.

---

> ### Author Response · Authors · 2020-11-16
> **Author Response - Part 1**
>
> We would like to thank you for your comments and suggestions. Below, you can find our responses to your reviews:
>
> Question 1: “In the mixture part, how do you train for the parameters of the mixture ? Do you have a kind of EM algorithm, or do you perform a gradient ascent?”
>
> Answer: In practice, the mixture weights are chosen to be uniform and mixture concentration parameters, i.e., $\kappa_{1}, …, \kappa_{k}$, are set to be equal to $\kappa$. It is because we want every component of the mixtures of vMF distributions to be treated similarly. In our model, the values of $k$, the number of components, and $\kappa$ will be tuned while the locations of vMF components will be optimized via the stochastic gradient ascent with the usage of the reparameterization trick.
>
> Question 2:  “In general, directions are drawn randomly for every batch of samples. What is the meaning of fixing in advance the number of projections, as done in Figure 3 and the related experiment ?”
>
> Answer: Thank you for your comment. For each batch of samples, the directions are drawn randomly. By fixing in advance the number of projections, we mean that the number of projections is the same for each batch of samples.
>
> Question 3: “The size of the mini-batch might have an impact on the estimation quality, that should be discussed.”
>
> Answer: In the revised version, we have conducted extra experiments to investigate the effect of minibatch’s size in Table 2 in Appendix E. We set the size of minibatch $\in$ { 10, 50,100, 150} in training DRAE, s-DRAE, ps-DRAE and we also adjust the number of epochs in each case to get the same number of iterations. From these experiments, we observe that the size of minibatch affects strongly the quality of DRAE, s-DRAE, and ps-DRAE. When the size is very small  ( = 10), all models perform poorly in both generation and reconstruction. On the other hand, a bigger minibatch size leads to better performance, namely, the FID score and reconstruction score decrease.

---

> ### Author Response · Authors · 2020-11-16
> **Author response - Part 2**
>
> Question 4: “If computing 1D FGW on mini batches, why not computing and comparing with the mini batch version of the original version of FGW ? From the paper, the batch size is 100, this would be solved very quickly by current FGW solvers.”
>
> Answer: Thank you for your insightful comment. There are two reasons that we have not compared our sliced-fused divergences with the original version of FGW.
>
> The first reason is that the FGW essentially aims to find an optimal transport plan to minimize the weighted average of Wasserstein distance and Gromov Wasserstein distance. Computing the FGW is as difficult as solving a quadratic programming problem, which is famous for its expensive complexity of at least the order $n^{4}$ [2] where $n$ is the number of components of each target probability measure. When the batch size is 100, the complexity of computing FGW will be at the order $100^4$, which is rather expensive. On the other hand, our sliced-fused divergences have the computational complexity of the order $n \log n$. It means that if we use the batch size 100, the complexity of the proposed sliced-fused divergences is at the order $100 \log 100$, which is 100^3 cheaper than that from FGW. Since we need to repeatedly compute either the original version of FGW or the proposed sliced-fused divergences during the training, we believe that using sliced-fused divergences gives us an edge over FGW in the computational time. However, we would like to remark that there is a trade-off between the computational time and accuracy here as the FGW functions directly in the original space of the target probability measures while the sliced-fused divergences are through projection steps. Needless to mention, in high dimension settings, sliced-fused divergences will not be able to keep all useful spatial information of the original measures. Therefore, the results from sliced-fused divergences will not be as good as those from FGW. Our proposed sliced-based divergences can be thought to try to balance the trade-off between computational time and the accuracy of the results by simultaneously searching for several informative directions for projection while making sure that the complexity of the proposed divergences still linear on the number of supports $n$ of the probability measures.
>
> The second reason that we have not compared our sliced-fused divergences to the original version of the FGW is because of the well-known curse of dimensionality with optimal transport. In Theorems 3, 6, 9 of our paper, we demonstrate that the proposed sliced-fused divergences, including SSFG, MSSFG, PSSFG, do not suffer from the curse of dimensionality. It essentially means that the sample complexity for these proposed divergences for estimating some unknown distribution does not increase with the dimension. On the other hand, the sample complexity of the optimal transport increases linearly with the dimension. Given that FGW is built upon optimal transport, it also suffers from the curse of dimensionality. As a consequence, the sliced-fused divergences will be statistically more efficient than FGW though they may not yield as good results as the FGW due to the limitation of the projection steps.
>
> We note in passing that projecting the probability measures into linear space of  higher dimensions than one (but still much smaller than the original dimension), which we refer to as subspace sliced-fused divergences, may be able to solve the issues with the original FGW. This idea was also advocated in the recently introduced subspace robust Wasserstein distance[3] or projected robust Wasserstein distance [4]. However, as these works pointed out, the computational complexity of these subspace distances (and similarly, subspace sliced-fused divergences) can still be very expensive (as it is equivalent to solving a Riemannian optimization objective problem efficiently). Therefore, better optimization schemes are still needed to develop to bring the subspace versions of FGW into the practical applications of deep learning.
>
> [2] G. Peyré, M. Cuturi, and J. Solomon. Gromov-wasserstein averaging of kernel and distance matrices. ICML, 2016.
>
> [3] F. Paty, and M. Cuturi. Subspace robust Wasserstein distances. ICML, 2019.
>
> [4] T. Lin, C. Fan, N. Ho, M. Cuturi, M. I. Jordan. Projection Robust Wasserstein Distance and Riemannian Optimization. NeurIPS, 2020.
>
> Question 5: “It is also a little bit strange that the original fused Gromov-Wasserstein paper is not credited in the paper: Titouan, Vayer, et al. "Optimal Transport for structured data with application on graphs." International Conference on Machine Learning. 2019”
>
> Answer:  Thank you for your reference. We have added this citation in the revised version of our paper.

---

> ### Author Response · Authors · 2020-11-16
> **Author response - Part 3**
>
> Question 6: “It is interesting to note that another paper in consideration for ICLR is developing a similar strategy in the case of the simple sliced Wasserstein distance: https://openreview.net/forum?id=QYjO70ACDK.”
>
> Answer: We would like to thank you for pointing out that paper. We would like to take this opportunity to explain the key difference between the two works. In the paper in https://openreview.net/forum?id=QYjO70ACDK,  the authors use an implicit distribution, which is later translated into a Borel family of measurable mappings from sphere to sphere and is approximated by a neural network, to cover important directions on the unit sphere. While this approach is efficient and interesting, the optimal measures can be a bit black-box to interpret. On the other hand, we approach the problem from a different angle, namely, we use mixtures of vMF or power spherical distributions as alternatives to the uniform distribution. Our proposed approaches encourage the samples to be concentrated around the centers of the mixtures, thereby reducing the unimportant directions drawn from the uniform distribution. Moreover, our proposed divergences can also interpolate between  the uniform distribution and Dirac distribution over the sphere (through the nice properties of vMF and PS distributions) while the approach in https://openreview.net/forum?id=QYjO70ACDK can generalize only the Dirac distribution.

---

### Official Review · AnonReviewer2 · 2020-10-31
**A novel method to improve relational regularized autoencoders**

**Rating:** 6
**Confidence:** 3

**Review:**

##########################################################################

Summary:

The paper presents a novel method to improve the relational regularized autoencoders. The proposed method is based on the new relational discrepancy which is called the spherical sliced fused Gromov Wasserstein (SSFG). It is seen that the SSFG is an extension of the sliced fused Gromov Wasserstein (SFG) and its max version. Two variants of the SSFG are also presented. Experiments suggest that the proposed autoencoders outperform some existing autoencoders in terms of generative performance in comparable computational time.


##########################################################################

Reasons for score:

Overall, I have a positive impression about the paper. I think that the proposed relational discrepancy using the von Mises--Fisher distribution is a reasonable extension of the SFG using the uniform distribution. My concern is whether the estimation of the tuning parameters of the proposed relational discrepancies could be computationally expensive in practice (see cons below). Hopefully the authors can address my concern in the rebuttal period.


##########################################################################

Pros:

(1) Applying the fact the von Mises--Fisher distribution is an extension of the uniform distribution and the Dirac distribution, the authors successfully presented an extended relational discrepancy of the SFG and its max version.

(2) The mixture spherical sliced fused Gromov Wasserstein (MSSFG) achieves an even better flexibility than the SSFG. I reckon that the MSSFG, which adopts a mixture of the von Mises--Fisher distributions, is a reasonable extension of the SSFG.

(3) The extensive experiments suggest that the proposed autoencoders show satisfactory performance in terms of FID scores and reconstruction losses and are not particularly expensive in terms of computational time.


##########################################################################

Cons:

(1) I wonder whether the estimation of the tuning parameters of the proposed relational discrepancies could be computationally expensive in practice. For example, the MSSFG requires the values of $k$, $\kappa_1,\ldots,\kappa_k, \alpha_1 , \ldots, \alpha_k $. I fear that the estimation of these tuning parameters requires expensive computational cost. In the experiments of the paper, only limited combinations of these tuning parameters are considered. I wonder whether these combinations really cover a sufficient area of the parameter space.

(2) The von Mises--Fisher distribution has the property that as the concentration parameter $\kappa$ increases, the concentration of the distribution monotonically increases. In particular, the von Mises--Fisher distribution tends to the Dirac distribution as $\kappa$ goes to infinity. However I am not sure this nice property also holds for the power distribution. If not, I wonder whether the power SSFG and power spherical DRAE have sufficient flexibility compared with the SSFG and spherical DRAE.


##########################################################################

Questions during rebuttal period:

Please address and clarify the cons above.

#########################################################################

Typo:

(1) p.1, abstract, last line: generation ,and reconstruction -> generation, and reconstruction

---

## Updates:

The authors have carefully responded to my comments. Their response addresses most of my concerns. I will keep my score high. I understand that the choice of the hyperparameters can be computationally heavy, but the authors have given an idea to solve this problem. It is good to find that the Power Spherical distribution also includes the Dirac distribution as a limiting case.

---

> ### Author Response · Authors · 2020-11-16
> **Author Response**
>
> Thank you for your time. We have revised our papers based on your comments. The changes are marked in blue color.
>
> Question 1: “I wonder whether the estimation of the tuning parameters of the proposed relational discrepancies could be computationally expensive in practice. For example, the MSSFG requires the values of k, κ1,…,κk,α1,…,αk. I fear that the estimation of these tuning parameters requires expensive computational cost. In the experiments of the paper, only limited combinations of these tuning parameters are considered. I wonder whether these combinations really cover a sufficient area of the parameter space.”
>
> Answer: In practice, we use uniform weights for {$\alpha_{1:k}$}  and the same value $\kappa$ for $\kappa_{1}, …, \kappa_{k}$ because we would like every component of the mixtures of vMF distributions to be treated similarly. Therefore, we eventually only need to tune for two hyperparameters $\kappa$ and $k$. In practice, we have searched only for $\kappa \in$ {1,5,10,50,100} and $k \in$ {2, 5, 10, 50} as we observed that the performance of MSSFG when $\kappa > 100$ is rather similar to when $\kappa = 100$ (similarly, the performance of MSSFG when $\kappa < 1$ is quite similar to when $\kappa = 1$). That's why we only consider $\kappa \in$ {1, 5, 10, 50, 100}. For the choice of the number of components $k$, we observe that $k \leq 50$ is well-suited for the current applications considered in the paper. However, we anticipate that for much larger-scale applications, a bigger $k$ should be chosen. One possible method for automatically choosing $k$ without tuning it is by using the idea from Bayesian nonparametrics, namely, we model the projections as Dirichlet process mixture models (cf. [1] and [2]) where the number of components $k$ grows automatically at certain rates (controlled by the hyperparameters in the Dirichlet process)  when the number of projections is increasing. We believe that it is an important direction to explore and we leave this direction for the future work.
>
> [1] T. S. Ferguson. A Bayesian Analysis of Some Nonparametric Problems. Annals of Statistics, 1973.
>
> [2] C. E. Antoniak. Mixtures of Dirichlet Processes with Applications to Bayesian Nonparametric Problems. Annals of Statistics, 1974.
>
> Question 2: “The von Mises--Fisher distribution has the property that as the concentration parameter κ  increases, the concentration of the distribution monotonically increases. In particular, the von Mises--Fisher distribution tends to the Dirac distribution as κ goes to infinity. However I am not sure this nice property also holds for the power distribution. If not, I wonder whether the power SSFG and power spherical DRAE have sufficient flexibility compared with the SSFG and spherical DRAE.”
>
> Answer: Thank you for your comment. In the revised version, we have proved that the Power Spherical distribution is also the interpolation between the uniform distribution and the Dirac distribution. This result is proved in Lemma 1 in Appendix D. Therefore, the power SSFG and power spherical DRAE are also as flexible as the SSFG and spherical DRAE.
>
> Question 3: “(1) p.1, abstract, last line: generation ,and reconstruction -> generation, and reconstruction.”
>
> Answer: We have fixed these typos in the revised version in blue color.

---

### Official Review · AnonReviewer4 · 2020-11-02
**A moderate Wasserstein paper that fills a gap in recent literature with a straightforward solution.**

**Rating:** 6
**Confidence:** 4

**Review:**

The paper proposes a new pseudo-distance called the spherical slices fused Gromov Wasserstein distance (SSFGW). It builds on top of the slices fused Gromov Wasserstein distance (SFGW) that takes a weighted combination of the Wasserstein distance and the Gromov Wasserstein distance on sliced spaces. The paper tackles the problem of solving for the best sampling directions of slicing. Existing approaches either assume uniform sampling strategies or a single direction that maximizes the discrepancy of two measures. This paper uses von Mises-Fisher measures (vMF) as the bridge to combine the advantages of the two. Changing the parameters of vMF is equivalent to (non-linearly) interpolating between a Dirac impulse and a uniform distribution, which converges to uniform-sliced Wasserstein and max-sliced Wasserstein, respectively. The authors provided the proof of its pseudo metric properties and upper bound and extended their SSFGW to mixtures of vFW, creating MSSFGW, and applied these variants to deterministic relational regularized autoencoder (DRAE). The results of comparing the DRAE equipped with proposed distance with other DRAEs demonstrate its superiority in stability and generative capacity.

Some comments and questions:

-- Where does von Mises-Fisher distribution come from?

If I understand it correctly, von Mises-Fisher is a variant of Gaussian on hyperspheres. The authors can consider citing existing works or proving some arguments of choosing vMF as an interpolation of a uniform distribution and a Dirac.

-- Please refine 2 Background.

Definition 2: is $I_v$ actually $I_d$ in $C_d(k)$?


"The vMF distribution provides a way to have concentrated weight on the most important directions and assigns less weight to further directions. Therefore, we gain a better representation of the discrepancy between probability measures."

Swap SFG and DRAE in the title of 2.1 to consist with the content of 2.1.


-- The introduction of power SSFG at the end of Section 3 seems abrupt.
If power SSFG performs better than SSFG, then what is the point of introducing vMF? Can we build mixtures of power SSFG? The paper does not mention an mps-DRAE.
On page 20 of the Appendix, at the top, it seems power SSFG is superior to SSFG in all desired aspects, but ps-DRAE underperforms s-DRAE, as shown in Table 1. Any explanation?


-- Empirical evidence of interpolating between a uniform and a Dirac
In Figure 12, abd, the graphs of s-DRAE cross the red baseline with a large momentum when $k$ decreases to 1, which suggests that it could go far away from the baseline when $k \rightarrow 0$ because when $k=1$, vFM is still very far from a uniform distribution. Please explain that.

-- A Wasserstein paper or a generative modeling paper?
The main contribution of the paper, which tackles the problem of generalizing sliced Wasserstein distance and max-sliced Wasserstein distance, is on the new pseudo metric but the paper only argues the contribution from the perspective of its power in generative modeling. I wonder if the authors ever considered evaluating the metric without add-ons like a neural network. Some typical areas of applications of the Wasserstein distances like color transfers, shape interpolation, rigid transformation since it is Gromov Wasserstein, etc? Or is the evaluation without an AE setup not necessary?

---------------------------
--AFTER REBUTTAL--
---------------------------
I appreciate the authors response. Most of my trivial comments and questions have been resolved. I stand by my initial rating. This is a solid paper. The authors clearly introduce the problem and develop a clear story with a straightforward solution. The paper ends with extensive experiments. My main concern remains: the contributions of the paper to the ML community is moderate because the story is very narrow. I think the idea can be substantially extended to solving the fundamental problems in sliced Wasserstein distances but I don't object acceptance of the paper in the current form. I recommend the authors incorporate suggestions from all the reviewers and polish the language especially Section 2 to make the paper more accessible for readers outside the sliced Wasserstein community. Thank you.

---

> ### Author Response · Authors · 2020-11-16
> **Author response - Part 1**
>
> Thank you for your time. We have revised our paper based on your comments. The changes are marked in blue color.
>
> Question 1: “Where does von Mises-Fisher distribution come from? If I understand it correctly, von Mises-Fisher is a variant of Gaussian on hyperspheres.”
>
> Answer: Thank you for your comment. There are two main motivations for why we use von Mises-Fisher distribution:
>
> First, by adjusting the concentration parameter of vMF distribution, we can interpolate between the uniform distribution on the unit sphere and the Dirac distribution (cf. the reference [0]). Therefore, it allows us to control distributing weight to the most important direction and other directions based on the geodesic distance on the sphere. Optimizing over the family of vMF distributions helps us to identify where the best direction is as well as how much weight we need to put there in comparison with other less important directions.
>
> Second, von Mises Fisher can be viewed as a good alternative to the Gaussian distribution on the sphere. It is well-known that mixtures of Gaussian distributions can approximate any distribution well (cf. the reference [1]). Therefore, mixtures of von Mises Fisher distributions can be viewed as a good alternative to the uniform distribution on the sphere. Furthermore, by using the finite mixture of von Mises Fisher distribution as the prior distribution on the projections, we can guarantee that the samples drawn from the mixture will concentrate around the centers/ locations of the mixture with high probability. Therefore, optimizing the centers/ locations of the mixture allows us to not only reduce the unimportant samples but also encourage these centers to be sufficiently separate from each other such that the finite mixture can cover the uniform distribution better.
>
> [0] S. Sra. Directional statistics in machine learning: a brief review. arXiv preprint arXiv:1605.00316,2016.
>
> [1] S. Ghosal and A. van der Vaart. Entropies and rates of convergence for maximum likelihood and Bayes estimation for mixtures of normal densities. Annals of Statistics, 1233–1263, 2001.
>
> Question 2: “The authors can consider citing existing works or proving some arguments of choosing vMF as an interpolation of a uniform distribution and a Dirac.”
>
> Answer: Thank you for your suggestion. We already cited references in the proof of Theorem 2 (namely, the reference [0] that we mentioned in our response to your first question) in the Appendix (page 12) to show the vMF is the interpolation between the uniform distribution and the Dirac distribution. Moreover, in the revised version, we also included Lemma 1 in Appendix D to show that Power Spherical distribution is also an interpolation between the uniform distribution and the Dirac distribution.
>
> Question 3: ”Please refine 2 Background. Definition 2: is  Iv  actually  Id in  Cd(k)? Swap SFG and DRAE in the title of 2.1 to consist with the content of 2.1.”
>
> Answer: We have fixed these issues in the revised version in blue color. Thank you for your comment.
>
> Question 4: “The introduction of power SSFG at the end of Section 3 seems abrupt. If power SSFG performs better than SSFG, then what is the point of introducing vMF?”
>
> Answer:  Thank you for your question. Power spherical distribution is a recently introduced distribution (cf. the reference (De Cao & Aziz, 2020). The Power Spherical distribution), so its statistical properties are not as well-studied as those of the vMF distribution, which is arguably the most well-known distribution on the unit sphere. Our main motivation for using power spherical distribution is from the computational side as the power spherical distribution has a faster sampling time than vMF distribution in high dimension settings. Furthermore, mixtures of vMF distributions can yield a good coverage of the uniform distribution over the unit sphere while it is unclear whether this property holds for power spherical distribution. Therefore, we think that it is necessary to include both the vMF and power spherical distributions in the paper since each distribution has its own merits in either statistical or computational front.

---

> ### Author Response · Authors · 2020-11-16
> **Author response - Part 2**
>
> Question 5: “On page 20 of the Appendix, at the top, it seems power SSFG is superior to SSFG in all desired aspects, but ps-DRAE underperforms s-DRAE, as shown in Table 1. Any explanation?”
>
> Answer: Thank you for your insightful observation. Let us take this opportunity to explain this possible confusing point. In fact, s-DRAE does not totally outperform ps-DRAE since ps-DRAE gives comparable results to s-DRAE with faster computational speed. Furthermore, in this autoencoder application, the latent space where SSFG and PSSFG are computed, is quite small (8 on MNIST dataset and 64 on CelebA dataset). Therefore, the vMF might not be strongly influenced by the dimension, thus, s-DRAE gives a very appealing performance.
>
> Question 6: “Can we build mixtures of power SSFG? The paper does not mention an mps-DRAE.”
>
> Answer: Thank you for your comment. In the revised version, we have included the mixture of power spherical and implemented the corresponding mixture power spherical sliced fused Gromov Wasserstein and its relational regularized autoencoder, named mps-DRAE. We have added these experiment results in Table 1, Figure 2, and Figures 6-11. Based on these results, mps-DRAE has comparable generative performance (in terms of FID score in both MNIST and CelebA) to ms-DRAE, but it has better computational speed than the ms-DRAE.
>
> Question 7: “Empirical evidence of interpolating between a uniform and a Dirac In Figure 12, abd, the graphs of s-DRAE cross the red baseline with a large momentum when k decreases to 1, which suggests that it could go far away from the baseline when  k→0, because when k=1, vFM is still very far from a uniform distribution. Please explain that.”
>
> Answer: Thank you for your comment. If $\kappa$ is small (less than one), the shapes of vMF and power spherical are not so different and close to the uniform distribution; therefore, the performance of s-DRAE and ps-DRAE will be stable and similar to DRAE. To demonstrate that,  we have included two new figures (Figures 13 and 16 in the Appendix) in the revised version to evaluate the performance of s-DRAE, ps-DRAE when $\kappa \in$ {0.001,0.01,0.1}. According to these figures, the performance of s-DRAE and s-DRAE at these values of $\kappa$ are very close to each other and are about equal to those of DRAE.
>
> Question 8: “A Wasserstein paper or a generative modeling paper? The main contribution of the paper, which tackles the problem of generalizing sliced Wasserstein distance and max-sliced Wasserstein distance, is on the new pseudo metric but the paper only argues the contribution from the perspective of its power in generative modeling. I wonder if the authors ever considered evaluating the metric without add-ons like a neural network. Some typical areas of applications of the Wasserstein distances like color transfers, shape interpolation, rigid transformation since it is Gromov Wasserstein, etc? Or is the evaluation without an AE setup not necessary?”
>
> Answer: Thank you for your suggestion. In the revised version, we have applied our methods to color image adaptation (color transfer), which uses the sliced-based optimal transport methods. We compare our techniques, which use the vMF distribution, power spherical distribution, and their mixtures to cover areas of informative projecting directions, with the uniform sampling of directions and the "max" direction approaches in Figure 21 in Appendix E.6. Based on the experiment results, transferred images from the uniform distribution are quite blurred and not catchy. "Max" direction often creates noises in the images (see first two examples in Figure 21). On the other hand, our methods, including vMF, PS, MvMF, and MPS, perform very well on color image adaptation, namely, the adapted images have similar color styles to the corresponding target images.

---

> > ### Comment · AnonReviewer4 · 2020-11-22
> > **Figure 21, color mismatch; Motivation**
> >
> > Thanks very much for adding the experiment. I think it is necessary to evaluate the metric without a neural network. And honestly even this experiment might not be enough. I could imagine there are many other aspects from which we can evaluate a newly proposed metric. Please put more thoughts there. No additional experiments needed.
> >
> > In figure 21, the orange color from the "suns" disappear in the transferred images. Any explanation?
> >
> > Another question, just to clarify for the purpose of review.
> > Improving the performance of SFG is the motivation of the work, but the paper is actually solving the underlying problem that sliced Wasserstein distances uniformly (or randomly sometimes in practice) project distributional data to 1D spaces, which diminishes its discrepancy capacity. That alone could qualify as a standalone paper and we can show its impact through many of its applications. Yet this paper finds itself in a specific focus on improving SFG and the story becomes extremely narrow as it is developing. Do the authors not think that the discussion in the main paper about the metric itself is not sufficient? Or did I understand the paper correctly in the current form?

---

> > > ### Author Response · Authors · 2020-11-23
> > > **Response to: Figure 21, color mismatch; Motivation**
> > >
> > > We would like to thank the reviewer for spending time reading our rebuttal and providing valuable feedbacks. Below, you can find our response with your questions:
> > >
> > > --- Question 1: “In figure 21, the orange color from the "suns" disappear in the transferred images. Any explanation?”
> > >
> > > Answer: Thank you for your question. Before answering your question, let us explain the procedure of how we perform color transfer, which we use from the paper [2]. First, we use K-means clustering to reduce the number of colors of the input and target images from 120000 to 3000, so the color might not be exactly 100%. After that, we sample directions from the sphere (with corresponding distributions of different methods). Then, with each direction, we project two compressed measures and then find the pixel-alignment between two projected measures. With each alignment, we create a transferred image with the pixel colors that are transported from the target images. Finally, we take the average of all transported images to get the final one.
> > >
> > > Going back to your question, we observe that the orange color from the suns is still in the transferred images though it is quite blurred and may look quite like yellow color. There are two reasons for this phenomenon:
> > >
> > > First reason: K-means clustering step uses the yellow color to represent the cluster that contains the orange pixels, which reduces the brightness of the orange pixels. This phenomenon happens because there are more yellow pixels than orange ones. We think that increasing the number of clusters of K-means may help but at the cost of slower computation.
> > >
> > > Second reason: The average of transferred images in the final step might reduce the brightness of the color. It is due to the difference between the 1D transport maps which are found by different projecting directions in the sliced distances. For example, in one map, a pixel is transferred to the orange color, however, in another map it is transferred to the blue color. Therefore, the average will not be orange anymore.
> > >
> > > Developing an efficient way to improve these two possible problems in the color transfer tasks is an important direction and we leave it for the future work. Finally, we have included these explanations in the revised version of the paper.
> > >
> > > [2] B. Muzellec and M. Cuturi. Subspace detours: Building transport plans that are optimal on subspace projections. NeurIPS, 2019.
> > >
> > >
> > > --- Question 2: “Another question, just to clarify for the purpose of review. Improving the performance of SFG is the motivation of the work, but the paper is actually solving the underlying problem that sliced Wasserstein distances uniformly (or randomly sometimes in practice) project distributional data to 1D spaces, which diminishes its discrepancy capacity. That alone could qualify as a standalone paper and we can show its impact through many of its applications. Yet this paper finds itself in a specific focus on improving SFG and the story becomes extremely narrow as it is developing. Do the authors not think that the discussion in the main paper about the metric itself is not sufficient? Or did I understand the paper correctly in the current form?”
> > >
> > > Answer: Thank you for your comment. Let us take this opportunity to clarify the confusing point about motivation that you mentioned. Our original motivation with the paper is to develop a better version of sliced-fused Gromov Wasserstein (SFG) to improve the performance of relational regularized autoencoders (RAE). It leads to the notion of (mixture) spherical sliced fused Gromov Wasserstein and its variants. We are aware that the idea can be extended to other applications as you mentioned. However, we feel that focusing on a particular application, which in this case the RAE, allows us to fully explore the potentials of the proposed sliced-fused Gromov-Wasserstein divergences in this application, namely, we are able to carry out quite comprehensive experiments with these divergences (e.g., sensitivity to hyper-parameters, quality of generative models, visualization of the latent spaces, etc.)  to investigate their benefits over the SFG within the RAE framework. Though the idea of extending the paper to other applications, such as color transfer, is appealing, we think that it will require several more extensive experiments (as the natures of the new applications are different from RAE)  to justify the benefits of proposed divergences over state-of-the-art distances in these applications. Given that the paper with just RAE is already quite long, we think that it is better to leave an extension of the proposed sliced-fused divergences to other applications in the future work.
> > >
> > > Finally, to avoid possible confusion about motivation as you mentioned, we have added a paragraph about the possible applicability of the proposed sliced-fused Gromov Wasserstein divergences to other applications in the introduction (in blue color).

---

### Author Response · Authors · 2020-11-24
**Final feedback before the discussion period ends**

Dear reviewers,

We would like to thank you again for spending your time evaluating our paper and reading the rebuttals.

Since the discussion period is going to send soon, we look forward to hearing your feedback about whether we have addressed your concerns in both the rebuttals and the revision of our paper. We would be happy to discuss if you have any other concerns.

Best,

Authors

---

### Decision · Program_Chairs · 2021-01-07
**Final Decision**

**Decision:**

Accept (Poster)

**Comment:**

This is a solid paper that proposes a new slicing approach to the fused Gromov Wasserstein distance using projection on directions sampled from  von-mises fisher direction, the location parameter of the von mises fisher is choosen to be maximally discriminating between the distribution
$(\max_{\epsilon}\mathbb{E}_{\theta|vMF(\theta|\epsilon,\kappa)}\beta W(\theta\mu,\theta\nu) +(1-\beta)GW(\theta\mu,\theta\nu))$, the new sliced distance is analyzed and extended to mixture of von mises distributions with $k$ locations or directions.  This contribution of the paper is of general interest beyond the application of the paper as mentioned by the reviewers.  Authors applies the new sliced Fused Gromov Wasserstein distance to relational auto encoders and show improvement.

The spherical slicing is original and new and of independent interest and the application is good as it pushes the boundary of relational auto encoders .Reviewers  and AC did not have any concerns with the paper and the rebuttal and revisions addressed all questions raised. Accept